# ADAPTIVE MLP PRUNING FOR LARGE VISION TRANSFORMERS

## ABSTRACT

Large vision transformers present impressive scalability, as their performance can be well improved with increased model capacity. Nevertheless, their cumbersome parameters results in exorbitant computational and memory demands. By analyzing prevalent transformer structures, we find that multilayer perceptron (MLP) modules constitute the largest share of the model's parameters.

In this paper, we propose an Adaptive MLP Pruning (AMP) method to substantially reduce the parameters of large vision transformers without obvious performance degradation. First, we adopt Taylor based method to evaluate neuron importance of MLP. However, the importance computation using one-hot cross entropy loss ignores the potential predictions on other categories, thus degrading the quality of the evaluated importance scores. To address this issue, we introduce label-free information entropy criterion to fully model the predictions of the original model for more accurate importance evaluation. Second, we rank the hidden neurons of MLP by the above importance scores and apply binary search algorithm to adaptively prune the ranked neurons according to the redundancy of different MLP modules, thereby avoiding the predefined compression ratio.

Experimental results on several state-of-the-art large vision transformers, including CLIP and DINOv2, demonstrate that our method achieves roughly 40% parameter and FLOPs reduction in a near lossless manner. Moreover, when the models are not finetuned after pruning, our method outperforms other pruning methods by significantly large margin. The source code and trained weights will be publicly available.

## 1 INTRODUCTION

Large transformers have demonstrated excellent scaling property on both computer vision and natural language processing, where their performance can be well improved as the model's capacity grows. However, their substantial computational and memory demands pose significant challenges for cost-effective deployment across a wide range of applications.

To reduce the size of large vision transformers, we analyze the parameter number of modules in vision transformers and find that MLP modules take dominant parameters of model. For example, in EVA-CLIP-E (Sun et al., 2023), MLP modules contain 81.1% parameters of the whole model. Hence, the pruning of MLP modules can significantly compress large vision transformers models.

Taylor based pruning methods present impressive performance on model compression. They evaluate the importance scores of weights according to the effect on model outputs after the elimination of the evaluated weights. Then, the least important weights of the model are pruned to compress the given model with minimal performance loss. Generally, these methods take one-hot cross entropy of outputs as the criterion for importance evaluation, where only the prediction corresponding to the given label contributes to importance evaluation. In other words, these methods inevitably ignore the other potential predictions during importance evaluation, thus hurting the fidelity of importance scores.

In this paper, we focus on the reduction of MLP modules in large vision transformers and propose an Adaptive MLP Pruning (AMP) method. First, we introduce a lable-free information entropy of model predictions as the general criterion for Taylor based pruning methods to evaluate importance

scores of MLP's hidden neurons. Different from one-hot cross entropy, our proposed information entropy criterion can fully model the possibility distribution of predictions, thus obtaining more accurate importance scores. Moreover, our proposed criterion doesn't rely on the loss function or extra modules adopted during the training of the original models, thereby enabling the compression of models, whose loss function or module weights are not fully published. For example, since the weights of DINO head module for the pretraining of DINOv2 (Oquab et al., 2023) aren't publicly available, the previous Taylor pruning methods can't be directly applied DINOv2.

Second, we rank the hidden neurons of MLP according to the importance scores obtained in the first stage. Then, we leverage binary search algorithm to determine the optimal number of pruned neurons for adaptive compression of MLP modules. During the search of optimal pruning, we evaluate the information entropy of the pruned model. If information entropy variation of model prediction after pruning exceeds the given threshold, we reduce the number of pruned neurons in the previous step. Otherwise, we further prune more hidden neurons, until the maximum number of search step is reached. In this manner, our method avoids the predefined pruning ratio as previous methods, thus achieving efficient and adaptive pruning of MLP modules for large vision transformers.

Finally, we perform knowledge distillation to recover the performance of the pruned model, where the original model serves as the teacher of the pruned one. Thanks to the structure affinity between the original model and the pruned one, the knowledge of the original model can be efficiently transferred to the pruned one for performance recover.

Our main contributions of our method can be summarized as below. **(1)** We introduce an information entropy criterion, not only providing more accurate importance scores for pruning, but also enabling label-free compression of large vision transformers without fully open codes or weights. **(2)** We propose an Adaptive MLP pruning method, which can effectively prune the redundant neurons of large vision transformers in an adaptive manner, thus avoid predefined pruning ratio as previous methods. **(3)** Only distilled on ImageNet-1K, our method achieves a near lossless acceleration of large vision transformers with roughly 40% parameter and FLOPs reduction. Moreover, when the pruned models are not finetuned for performance recover, our method significantly outperforms other pruning methods by a large margin.

## 2 RELATED WORK

### 2.1 MODEL PRUNING

To reduce the cost of vision transformer, model pruning methods are proposed to compress multi-head self-attention modules or multilayer perceptron modules. The core idea of these methods is to evaluate the importance of model weights and prune the least important weights, thus compressing models with less performance loss.

Magnitude-based methods (Han et al., 2015; Li et al., 2017; Liu et al., 2017) measure the importance score of weights by their magnitudes and the weights with large magnitude are regarded as the more important ones for final predictions. ViT-Slim (Chavan et al., 2022) introduces a learnable $\ell_1$ sparsity constraint as the global importance score in the continuous searching space to search an optimal ViT sub-structure for efficient inference. DIMAP (He & Zhou, 2024) evaluate the contribution of local weights by their information distortion to prune models without dependence on the input data.

Attention-based methods mine important weights by the attention scores obtained from multi-head self-attention modules. SNP (Shim et al., 2024) prunes graphically connected query and key layers with the least informative attention scores, while keeping the overall attention scores.

Taylor pruning methods (Molchanov et al., 2017; 2019) evaluate the weight or neuron importance by approximating the change of loss function after pruning using Taylor expansion. VTC-LFC (Wang et al., 2022) adopts low-frequency sensitivity metric for Taylor expansion to evaluate importance scores for pruning. SAViT (Zheng et al., 2022) introduces joint importance evaluation across all component for Taylor pruning to achieve a more balanced parameter reduction. NViT (Yang et al., 2023) proposes a Hessian-based pruning criterion for global model pruning.

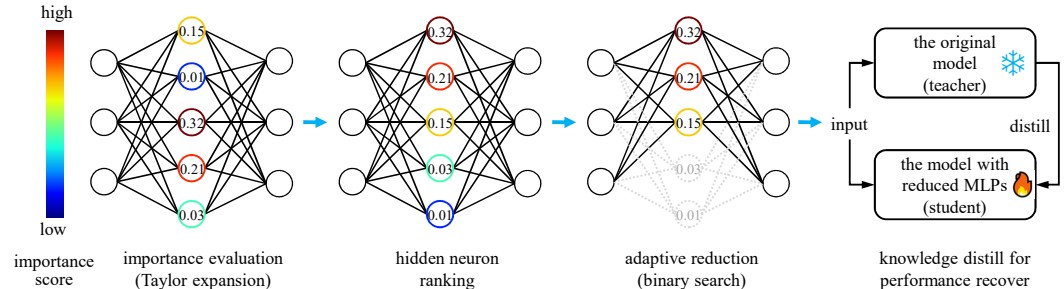

Figure 1: The overview of the proposed method. First, the importance scores of hidden neurons are evaluated by Taylor based method. Then, we rank the hidden neurons by the obtained importance scores. Afterwards, we conduct binary search to adaptively prune the hidden neurons for MLP modules in transformer. Finally, the pruned model is guided by the original model using knowledge distillation to recover performance.

## 2.2 TOKEN REDUCTION

To accelerate the inference of vision transformer, another alternative solution is to reduce the number of tokens fed into the model. There exist two research lines for token reduction, including token pruning and token merging.

Token pruning methods eliminate unimportant tokens of input sequence to reduce the cost of model inference. DynamicViT (Rao et al., 2021) proposes an attention masking strategy to prune the redundant token by blocking its interactions with other tokens. AdaViT (Meng et al., 2022) dynamically applies the patch tokens, heads and transformer layer according to the input images, thus improving the inference efficiency. A-ViT (Yin et al., 2022) exploits adaptive halting mechanism to prune the non-discriminative tokens at different layers of transformer model. LRP (Luo et al., 2024) prunes unimportant tokens according to semantic density score of each patch, which is measured by the variation between reconstructions with and without this patch.

Token merging methods fuse several redundant tokens into one, thereby reducing the number of tokens for efficient inference. ToMe (Bolya et al., 2023) exploits bipartite soft matching algorithm to efficiently merge the most similar tokens for token reduction. TPS (Wei et al., 2023) leverages unidirectional nearest-neighbor matching and similarity-based fusing steps to squeeze the number of tokens. BAT (Long et al., 2023) merges similar inattentive tokens and match attentive tokens to maximize the diversity of tokens after token reduction. STViT (Chang et al., 2023) constructs several cluster centers to represent the whole token sequence for inference acceleration.

Our method focuses on parameter reduction of large vision transformers and is fully compatible with the above token reduction methods. The combination of our method and token reduction methods can further improve the inference efficiency of large vision transformers.

## 3 THE PROPOSED METHOD

### 3.1 OVERVIEW

To effectively compress large vision transformers, we focus on the reduction of parameter-intensive MLP modules in the transformer architecture. The overview of our proposed method is depicted in Figure 1. First, we evaluate importance scores of hidden neurons in MLP module according to the sensitivity of model predictions to the elimination of each neuron. Then, we rank the hidden neurons by the obtained importance scores to determine the order of neuron pruning, thus minimizing the performance drop after compression. Afterward, we conduct binary search algorithm on the sorted neurons to adaptively prune redundant hidden neurons in MLP module, until the tolerance error between the pruned model and original one is reached. Finally, we implement knowledge distillation between the original model and the pruned one to guide the performance recover of the compressed model. Since only hidden layers of MLP modules are pruned, the output dimension of the pruned model is identical to the original one. Hence, knowledge distillation between the original model and the pruned one can be directly applied without any additional alignment modules.

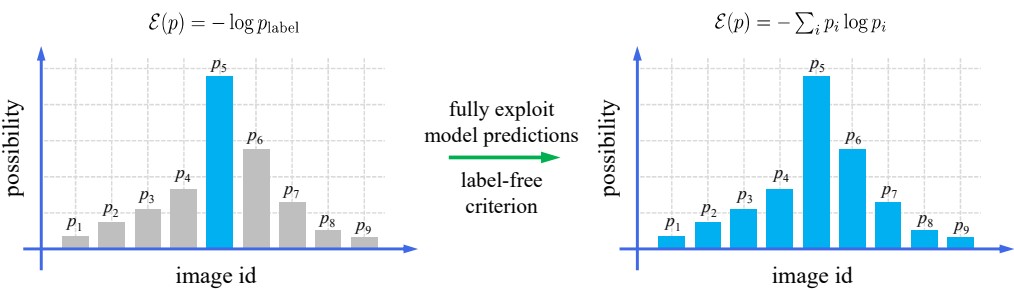

(a) one-hot cross entropy for importance evaluation      (b) information entropy for importance evaluation

Figure 2: One-hot cross entropy vs information entropy for neuron importance evaluation. Our proposed information entropy exploits all predictions of the model for more accurate importance evaluation.

### 3.2 PRELIMINARIES OF NEURON IMPORTANCE EVALUATION

The goal of model compression is to minimize the variance on model prediction after pruning, thus reducing the performance drop. Let $\mathcal{C}(\mathcal{D}, \mathcal{W})$ denotes the prediction criterion of the model, where $\mathcal{D}$ and $\mathcal{W}$ represent the dataset and the parameters of the model, respectively. Since we focus on neuron pruning (or structural pruning) in this paper, we express $\mathcal{C}(\mathcal{D}, \mathcal{W})$ as $\mathcal{C}(\mathcal{H})$ for convenience, where $\mathcal{H} = \{h_k\}_k$ refers to hidden feature set obtained by feeding dataset $\mathcal{D}$ into the model with parameters $\mathcal{W}$. The importance of $k$-th hidden neuron can be measured by the variance of $\mathcal{C}(\mathcal{H})$ as

$$\Delta \mathcal{C}_k = \mathcal{C}(\mathcal{H}_{h_k = \hat{h}_k}) - \mathcal{C}(\mathcal{H}_{h_k = 0}), \tag{1}$$

where $\hat{h}_k$ denotes the feature value of neuron $h_k$ before pruning; $h_k = 0$ refers to the pruning of $k$-th neuron; $\mathcal{H}_{h_k = \hat{h}_k}$ means that only the value of variance $h_k$ in $\mathcal{H}$ is set to $\hat{h}_k$.

To efficiently evaluate importance criterion in Eq. 1, we expand $\mathcal{C}(\mathcal{H})$ at point $h_k = \hat{h}_k$ using Taylor expansion (Molchanov et al., 2017) as

$$\mathcal{C}(\mathcal{H}) = \mathcal{C}(\mathcal{H}_{h_k = \hat{h}_k}) + \nabla_{\hat{h}_k}\mathcal{C} \cdot (h_k - \hat{h}_k) + R(h_k), \tag{2}$$

where $\nabla_{\hat{h}_k}\mathcal{C}$ denotes the gradient of $\mathcal{C}(\mathcal{H})$ w.r.t $h_k$ at point $h_k = \hat{h}_k$ and $R(h_k)$ refers to the first order remainder. Combining Eq. 1 and Eq. 2, we can obtain the importance score of the $k$-th neuron as

$$\Delta \mathcal{C}_k = \mathcal{C}(\mathcal{H}_{h_k = \hat{h}_k}) - \mathcal{C}(\mathcal{H}_{h_k = 0}) = \hat{h}_k \cdot \nabla_{\hat{h}_k}\mathcal{C} - R(h_k) \approx \hat{h}_k \cdot \nabla_{\hat{h}_k}\mathcal{C}, \tag{3}$$

where $R(h_k)$ is omitted for approximation.

For the sequence with $N$ tokens in large vision transformers, we evaluate the importance score of $k$-th hidden neuron in MLP module by

$$\mathcal{I}_k = \left| \sum_{n=1}^{N} \Delta \mathcal{C}_k^{(n)} \right| = \left| \sum_{n=1}^{N} \hat{h}_k^{(n)} \cdot \nabla_{\hat{h}_k^{(n)}}\mathcal{C} \right|, \tag{4}$$

where all $k$-th hidden neurons are summed over token sequence for multi-variate function $\mathcal{C}(\mathcal{H})$ and $\Delta \mathcal{C}_k^{(n)}$ denotes the importance of $k$-th neuron from $n$-th token in sequence.

### 3.3 INFORMATION ENTROPY FOR NEURON IMPORTANCE EVALUATION

Generally, the previous Taylor based methods (Molchanov et al., 2017) take one-hot cross entropy loss as the criterion $\mathcal{C}(\mathcal{H})$ for the computation of importance score $\mathcal{I}_k$ to measure the sensitivity of model performance after pruning. Nevertheless, as Fig. 2 (a), one-hot cross entropy criterion only takes the prediction probability corresponding to label into account, while other prediction probabilities are fully ignored, thereby missing key information during importance evaluation.

In this section, we introduce information entropy as the criterion for importance evaluation as Fig. 2 (b), where all prediction possibilities are exploited for more accurate importance scores. However,

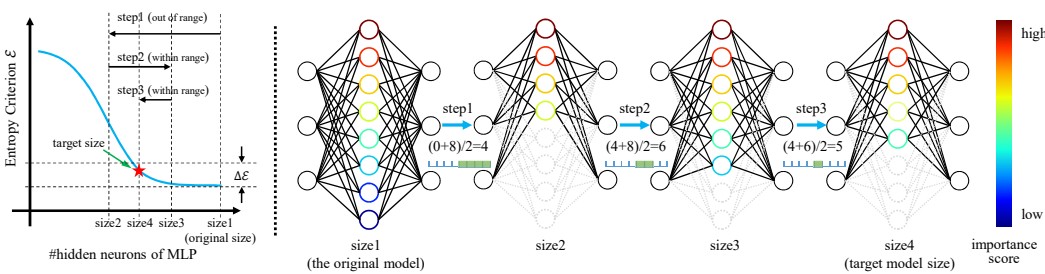

Figure 3: Adaptive MLP Pruning. In each pruning step, we conduct binary search algorithm to adaptively reduce the search range of optimal hidden size into half according to information entropy $\mathcal{E}$, until the maximum pruning step number reaches. If the increment of information entropy within the range of $\Delta\mathcal{E}$, we further prune the hidden neurons of MLP. Otherwise, we reduce the number of pruned neurons in the previous step.

the original prediction possibilities of large vision transformers are not always available. For example, DINOv2 series models (Oquab et al., 2023) only provides the weights of backbone, while the module weights for prediction possibilities are not published.

To this end, we introduce a general solution based on inter-instance similarity to compute the prediction possibilities without dependency on extra module or the original loss function. Specifically, we obtain inter-instance similarity matrix $s \in \mathbb{R}^{B \times B}$ between $B$ image representations in mini-batch. The similarity between the $i$-th image and the $j$-th one can be computed by

$$s_{ij} = \frac{z_i^{\text{cls}} \cdot z_j^{\text{cls}}}{\|z_i^{\text{cls}}\| \cdot \|z_j^{\text{cls}}\|}, \tag{5}$$

where $z_i^{\text{cls}}$ stands for the representation of the $i$-th image output by the last block of transformer model. Then, we apply softmax operation on similarity matrix $s$ to obtain prediction possibility matrix $p \in \mathbb{R}^{B \times B}$ by

$$p_{ij} = \frac{\exp\left(s_{ij}/\tau\right)}{\sum_{j'=1}^{B} \exp\left(s_{ij'}/\tau\right)}, \tag{6}$$

where $\tau$ is a temperature coefficient to scale the range of $s_{ij}$ from $[-1, 1]$ to $[-1/\tau, 1/\tau]$. Finally, we obtain the information entropy criterion as below

$$\mathcal{E} = -\frac{1}{B} \sum_{i=1}^{B} \sum_{j=1}^{B} p_{ij} \cdot \log p_{ij}. \tag{7}$$

Compared to cross entropy criterion, our proposed information criterion has the following advantages. First, our criterion doesn't rely on the loss function for the training of the original model, thus unifying the importance evaluation of different models. Second, our criterion doesn't require labeled dataset for importance evaluation. Third, our criterion enables the importance evaluation without additional modules, e.g. DINO head module for DINOv2 and text encoder for CLIP, thus improving the evaluation efficiency.

### 3.4 ADAPTIVE MLP PRUNING

As shown in Fig. 3, the value of our proposed information entropy criterion decreases with the increment of hidden size in MLP. [1] In other words, the prediction uncertainty of model is reduced when the capacity of model is improved. We set an increment entropy threshold $\Delta\mathcal{E}$ after pruning to keep performance degradation within acceptable range. Let $M_0$ denotes the hidden size of MLP in the original model, where all hidden sizes of MLP modules in one transformer model are identical.

Our MLP pruning task can be formalized as a problem to search an optimal hidden size in the range of $[0, M_0]$. We set the search range to $[M_{\min}, M_{\max}]$, where $M_{\min}$ and $M_{\max}$ are initialized to 0 and

---

[1]We visualize this relation on OpenCLIP-g model in the appendix.

---

**Algorithm 1** Adaptive MLP Pruning

**Input:** The hidden size of the original model $M_0$; iteration number $t_{\max}$;

**Output:** The hidden sizes of MLPs after pruning $\{M_{\mathrm{res}}^{(l)}\}_{l=1}^L$

1: Evaluate the entropy of the model $\mathcal{E}_0^{(L)}$ on dataset $\mathcal{D}_{\mathrm{prune}}$;
2: **for** block $l = L$ to 1 **do**
3:     Initialize $M_{\min} = 0$ and $M_{\max} = M_0$;
4:     Initialize $M_{\mathrm{res}}^{(l)} = M_0$ and $\mathcal{E}_{\mathrm{res}}^{(l)} = \mathcal{E}_0^{(l)}$;
5:     **for** $t = 1$ to $t_{\max}$ **do**
6:        Obtain hidden size after pruning $M_t^{(l)} = \frac{M_{\min} + M_{\max}}{2}$;
7:        Evaluate the entropy of these pruned model $\mathcal{E}_t^{(l)}$ on dataset $\mathcal{D}_{\mathrm{prune}}$;
8:        **if** $\mathcal{E}_t^{(l)} - \mathcal{E}_0^{(l)} < \Delta\mathcal{E}$ **then**
9:           $M_{\max} = M_t^{(l)}$
10:          $M_{\mathrm{res}}^{(l)} = M_t^{(l)}$
11:          $\mathcal{E}_{\mathrm{res}}^{(l)} = \mathcal{E}_t^{(l)}$
12:        **else**
13:          $M_{\min} = M_t$
14:        **end if**
15:     **end for**
16:     $\mathcal{E}_0^{(l-1)} = \mathcal{E}_{\mathrm{res}}^{(l)}$
17: **end for**

---

$M_0$, respectively. To efficiently search the optimal hidden size, we follow the idea of binary search algorithm to evaluate information entropy $\mathcal{E}_t^{(l)}$ on small dataset $\mathcal{D}_{\mathrm{prune}}$ when the hidden size of block $l$ is reduced to $M_t^{(l)} = \frac{M_{\min} + M_{\max}}{2}$ at the search step $t$. If $\mathcal{E}_t^{(l)} - \mathcal{E}_0^{(l)} < \Delta\mathcal{E}$, we update the search range from $[M_{\min}, M_{\max}]$ to $[M_{\min}, M_t^{(l)}]$ for further pruning, where $\mathcal{E}_0^{(l)}$ denotes entropy before the pruning of block $l$ and $\Delta\mathcal{E}$ indicates the threshold of entropy variance. Otherwise, the search range is updated from $[M_{\min}, M_{\max}]$ to $[M_t^{(l)}, M_{\max}]$ to reduce the number of pruned neurons in the previous step. After each search step, the size of search range is reduced to half of the original one, until the maximum search step $t_{\max}$ is reached or the size of search range is reduced to 1. The overall algorithm is described in Algorithm 1.

### 3.5 KNOWLEDGE DISTILLATION

In this section, we take the original model as the teacher to guide the performance recover of the model with reduced MLP module. The outputs of teacher's last transformer block can be represented as $z^{\mathrm{cls}}$ and $z^{\mathrm{patch}}$, where $z^{\mathrm{cls}} \in \mathbb{R}^C$ and $z^{\mathrm{patch}} \in \mathbb{R}^{N \times C}$ are the embeddings of class token and patch tokens, respectively. Similarly, the outputs of student's last block are $\hat{z}^{\mathrm{cls}}$ and $\hat{z}^{\mathrm{patch}}$, whose dimension sizes are identical to the ones of $z^{\mathrm{cls}}$ and $z^{\mathrm{patch}}$.

To recover the performance of the pruned model, we conduct knowledge distillation using mean squared error loss on class token and patch tokens as

$$\mathcal{L}_{\mathrm{distill}} = \frac{1}{C}\|z^{\mathrm{cls}} - \hat{z}^{\mathrm{cls}}\|^2 + \frac{1}{N \cdot C}\|z^{\mathrm{patch}} - \hat{z}^{\mathrm{patch}}\|^2. \tag{8}$$

Due to the consistent dimension and the weight affinity between the original model and the pruned one, the model pruned by our method can efficiently transfer knowledge from the original model.

## 4 EXPERIMENTS

### 4.1 EXPERIMENTAL SETTINGS

We conduct knowledge distillation on training set of ImageNet-1K (Russakovsky et al., 2015) without labels, which contains only 0.06% data of LAION-2B (Schuhmann et al., 2022). We randomly

Table 1: The performance comparison on zero-shot image classification tasks of various ImageNet variants and ObjectNet. "#Params" denotes the number of vision encoder's parameters, excluding the parameters of text encoder. Throughout is averaged over 10 runs on single A6000 GPU with batch size of 1000. "prune" and "distill" indicate the pruned model without finetuning and the distilled one, respectively.

| Method | #Params | FLOPs | Throughout | IN-1K | IN-Adv | IN-R | IN-V2 | IN-Ske | ObjectNet | Avg. Acc. |
|---|---|---|---|---|---|---|---|---|---|---|
| OpenCLIP-g (Cherti et al., 2023) | 1.01B | 0.52T | 150.8 imgs/s | **78.5%** | **60.8%** | 90.2% | **71.7%** | 67.5% | 69.2% | 73.0% |
| OpenCLIP-g (ours, prune) | **0.62B** | **0.32T** | **222.3 imgs/s** | 65.0% | 34.0% | 74.4% | 57.7% | 43.9% | 47.9% | 53.8% |
| OpenCLIP-g (ours, distill) | **0.62B** | **0.32T** | **222.3 imgs/s** | 78.4% | **60.8%** | **90.5%** | 71.5% | **67.9%** | **69.4%** | **73.1%** |
| OpenCLIP-G (Wortsman, 2023) | 1.84B | 0.95T | 90.6 imgs/s | **80.1%** | **69.3%** | **92.1%** | **73.6%** | **68.9%** | **73.0%** | **76.2%** |
| OpenCLIP-G (ours, prune) | **1.17B** | **0.60T** | **135.8 imgs/s** | 70.2% | 37.1% | 78.3% | 62.7% | 50.9% | 51.1% | 58.4% |
| OpenCLIP-G (ours, distill) | **1.17B** | **0.60T** | **135.8 imgs/s** | **80.1%** | 68.4% | 91.9% | 73.4% | **68.9%** | 72.8% | 75.9% |
| EVA-CLIP-E (Sun et al., 2023) | 4.35B | 2.23T | 43.1 imgs/s | **82.0%** | 82.1% | 94.5% | **75.7%** | 71.6% | **79.6%** | 80.9% |
| EVA-CLIP-E (ours, prune) | **2.50B** | **1.28T** | **69.5 imgs/s** | 59.2% | 18.6% | 59.8% | 51.3% | 34.9% | 40.6% | 44.1% |
| EVA-CLIP-E (ours, distill) | **2.50B** | **1.28T** | **69.5 imgs/s** | **82.0%** | **82.3%** | **94.6%** | 75.1% | **71.7%** | **79.6%** | **81.0%** |
| EVA-CLIP-8B (Sun et al., 2024) | 7.53B | 3.86T | 20.8 imgs/s | 83.5% | 85.2% | **95.3%** | **77.7%** | **74.3%** | 81.2% | **82.9%** |
| EVA-CLIP-8B (ours, prune) | **4.59B** | **2.35T** | **30.1 imgs/s** | 61.7% | 49.9% | 66.5% | 53.1% | 34.9% | 47.9% | 52.3% |
| EVA-CLIP-8B (ours, distill) | **4.59B** | **2.35T** | **30.1 imgs/s** | **83.7%** | **85.6%** | **95.3%** | **77.7%** | 74.0% | **81.3%** | **82.9%** |

sample $50,000$ images from training set of ImageNet-1K as the dataset for binary search based pruning, namely $\mathcal{D}_{\text{prune}}$. All images for distillation and evaluation are resized into $224 \times 224$.

We evaluate our method on several popular benchmarks. For CLIP-style models, we evaluate the models on zero-shot image classification of various ImageNet variants (including ImageNet-1K, ImageNet-V2 (Recht et al., 2019), ImageNet-Adv (Hendrycks et al., 2021b), ImageNet-R (Hendrycks et al., 2021a) and ImageNet-Sketch (Wang et al., 2019)) and ObjectNet (Barbu et al., 2019) using CLIP benchmark (LAION-AI, 2023). To further validate the effectiveness of our method, we also estimate our method on zero-shot image and text retrieval tasks of Flickr30K (Young et al., 2014) and COCO (Lin et al., 2014). Additionally, all text encoder of CLIP-style models are fixed without pruning or finetuning. For DINOv2-g, we evaluate its performance model on ImageNet-1K using kNN evaluation protocol. Moreover, we also evaluate CLIP-style models using kNN evaluation protocol.

All models are trained on GPU servers with $8\times$ A6000 GPUs for 10 epochs, including the first 1 epoch for warming-up. We adopt AdamW (Loshchilov, 2017) optimizer with bfloat16 precision for model training. The learning rate follows a cosine schedule from lr to zero, where lr = base_lr $\times$ batch_size / 256. The base learning rates and batch sizes for different models are listed in the appendix. We set the number of step for binary pruning $t_{\max}$ to 6. Other pruning hyper-parameters, including $\Delta\mathcal{E}$ and temperature coefficient $\tau$, are set according to the backbones of pruned models and can be found in the appendix.

## 4.2 Zero-Shot Image Classification

To validate the effectiveness of our method, we prune the state-of-the-art CLIP models and then fine-tune the pruned models by knowledge distillation. Both pruned and distilled models are evaluated on zero-shot classification tasks of various ImageNet-1K variants and ObjectNet datasets. The results reported in Table 1 indicate that our method achieves about 40% parameter and FLOPs reduction for all models. We also report image throughout of the pruned models to evaluate their performance in reality. All models pruned by our method accomplishes roughly $1.5\times$ inference acceleration. In spite of large parameter reduction without finetuning, the models pruned by our method maintain a very promising performance on zero-shot classification tasks. When finetuned using knowledge distillation, our pruned models can well recover the performance as their original versions, respectively. In some cases, the distilled models, including OpenCLIP-g (ours, distill) and EVA-CLIP-E (ours, distill), even slightly outperform the original models.

## 4.3 Zero-Shot Retrieval

We further evaluate our pruned models on zero-shot retrieval tasks of Flickr30K and COCO datasets, including zero-shot text retrieval and zero-shot image retrieval tasks. The experimental results in Table 2 demonstrate that our distilled models consistently achieve comparable performance as the corresponding original models. For global metric, mean recall (MR), our distilled models, including OpenCLIP-g (ours, distill), EVA-CLIP-E (ours, distill) and EVA-CLIP-8B (ours, distill), even sur-

Table 2: The performance comparison on zero-shot retrieval task of Flickr30K and COCO datasets. "R@K" is short for Recall@K. "MR" is short for mean recall, which is average value of all recall metrics.

| Method | #Params | Zero-Shot Text Retrieval (text → image) | | | | | | Zero-Shot Image Retrieval (image → text) | | | | | | MR |
| | | Flickr30K | | | COCO | | | Flickr30K | | | COCO | | | |
| | | R@1 | R@5 | R@10 | R@1 | R@5 | R@10 | R@1 | R@5 | R@10 | R@1 | R@5 | R@10 | |
| OpenCLIP-g (Cherti et al., 2023) | 1.01B | 91.5 | 98.9 | 99.5 | **66.4** | 86.0 | 91.8 | **77.5** | **94.1** | **96.7** | 48.7 | 73.2 | 81.4 | 83.8 |
| OpenCLIP-g (ours, prune) | **0.62B** | 83.4 | 95.7 | 97.7 | 56.6 | 79.3 | 86.3 | 69.5 | 88.8 | 93.2 | 41.6 | 66.6 | 76.3 | 77.9 |
| OpenCLIP-g (ours, distill) | **0.62B** | 91.6 | **99.1** | **99.6** | 66.2 | **86.5** | **91.9** | **77.5** | **94.1** | **96.7** | 49.3 | 73.9 | 82.0 | **84.0** |
| OpenCLIP-G (Wortsman, 2023) | 1.84B | **92.6** | 99.4 | **99.8** | 66.9 | **87.2** | **92.8** | 79.8 | **95.1** | 97.0 | 51.3 | 74.8 | 82.9 | **85.0** |
| OpenCLIP-G (ours, prune) | **1.17B** | 87.8 | 97.9 | 99.2 | 60.7 | 83.0 | 89.9 | 72.9 | 91.2 | 94.8 | 44.2 | 69.2 | 78.2 | 80.8 |
| OpenCLIP-G (ours, distill) | **1.17B** | 92.0 | **99.6** | **99.8** | **67.1** | 87.0 | 92.7 | **79.9** | 95.2 | **97.1** | 51.5 | 75.3 | 83.2 | **85.0** |
| EVA-CLIP-E (Sun et al., 2023) | 4.35B | **94.9** | **99.3** | 99.7 | **68.8** | 87.8 | 93.0 | 78.9 | 94.4 | **97.1** | 51.0 | 74.8 | 82.7 | 85.2 |
| EVA-CLIP-E (ours, prune) | **2.50B** | 68.8 | 89.3 | 93.8 | 41.7 | 66.2 | 76.6 | 62.0 | 84.5 | 90.0 | 35.4 | 60.3 | 70.8 | 70.0 |
| EVA-CLIP-E (ours, distill) | **2.50B** | 94.4 | **99.3** | **99.8** | 68.6 | **87.9** | **93.2** | **79.6** | **94.5** | 96.7 | 51.3 | 74.9 | 82.8 | **85.3** |
| EVA-CLIP-8B (Sun et al., 2024) | 7.53B | **94.4** | 99.4 | 99.7 | 69.6 | 88.6 | 93.2 | 80.9 | 95.3 | 97.4 | 51.7 | 75.0 | 82.7 | 85.7 |
| EVA-CLIP-8B (ours, prune) | **4.59B** | 73.5 | 93.9 | 96.9 | 40.5 | 63.3 | 73.4 | 67.9 | 88.6 | 93.0 | 38.0 | 63.3 | 73.4 | 72.2 |
| EVA-CLIP-8B (ours, distill) | **4.59B** | 94.3 | **99.6** | **99.8** | **70.2** | **88.8** | **93.4** | **81.5** | **95.4** | **97.6** | **52.8** | **75.8** | **83.5** | **86.1** |

pass the original models with significantly fewer parameters. Especially, our distilled EVA-CLIP-8B achieves 0.4% MR gain over the original model. These results indeed support that our method can effectively reduce redundant parameters of MLP modules in large vision transformers in a lossless manner.

## 4.4 COMPARISON TO OTHER PRUNING METHODS

To evaluate the superiority of our method, we also compare our method with other state-of-the-art pruning methods. For random, $\ell_2$ and SAViT pruning, the hidden sizes of MLP module are pruned to 2645 for OpenCLIP-g and 7358 for EVA-CLIP-E, thus containing the equal parameter number as our pruned models for fair comparison. For Taylor pruning and NViT, we adopt adaptive pruning with consistent model

Table 3: Performance comparison of different pruning strategies on average zero-shot classification accuracy of ImageNet variants and ObjectNet. "original" denotes the original model without compression.

| Method | OpenCLIP-g | | EVA-CLIP-E | |
| | prune | distill | prune | distill |
| original | - | 73.0% | - | 80.9% |
| random pruning | 0.4% | 67.4% | 0.4% | 75.8% |
| $\ell_2$ norm pruning | 1.6% | 69.3% | 0.7% | 78.3% |
| Taylor pruning (Molchanov et al., 2017) | 9.6% | 70.6% | 1.5% | 79.1% |
| SAViT (Zheng et al., 2022) | 10.3% | 70.9% | 2.1% | 79.3% |
| NViT (Yang et al., 2023) | 11.1% | 71.0% | 2.2% | 79.6% |
| AMR (ours) | **53.8%** | **73.1%** | **44.1%** | **81.0%** |

parameters after pruning as our pruned model. As shown in Table 3, our pruned models outperform other pruning methods by a large margin, such as 42.7% performance gain on OpenCLIP-g, when the pruned models are not finetuned. When all pruned models are finetuned using knowledge distillation, our method consistently outperforms other methods. For example, our distilled EVA-CLIP-E outperforms the second best method NViT by 2.1% average zero-shot classification accuracy. The experimental results indeed support the superiority of our proposed method.

## 4.5 KNN EVALUATION

For more comprehensive comparison, we further evaluate our method on ImageNet-1K using kNN evaluation protocol, which is also applicable to pure vision transformer, such as DINOv2-g. As the results in Table 4, our distilled models achieve comparable performance as the original models with significantly fewer parameters, even slightly superior to the original ones in some cases. For example, EVA-CLIP-E (ours, distill) improves kNN accuracy from 85.8% to 85.9%, while only leveraging 57.5% parameters of the original one. For pure vision transformer, the per-

Table 4: The performance comparison on ImageNet-1K using kNN evaluation protocol. The outputs of the last transformer block are used to evaluate the kNN performance.

| Method | #Params | FLOPs | kNN (%) |
|---|---|---|---|
| OpenCLIP-g (Cherti et al., 2023) | 1.01B | 0.52T | **81.7** |
| OpenCLIP-g (ours, prune) | **0.62B** | **0.32T** | 72.8 |
| OpenCLIP-g (ours, distill) | **0.62B** | **0.32T** | **81.7** |
| OpenCLIP-G (Wortsman, 2023) | 1.84B | 0.95T | **82.9** |
| OpenCLIP-G (ours, prune) | **1.17B** | **0.60T** | 74.7 |
| OpenCLIP-G (ours, distill) | **1.17B** | **0.60T** | 82.8 |
| EVA-CLIP-E (Sun et al., 2023) | 4.35B | 2.23T | 85.8 |
| EVA-CLIP-E (ours, prune) | **2.50B** | **1.28T** | 72.2 |
| EVA-CLIP-E (ours, distill) | **2.50B** | **1.28T** | **85.9** |
| EVA-CLIP-8B (Sun et al., 2024) | 7.53B | 3.86T | 86.0 |
| EVA-CLIP-8B (ours, prune) | **4.59B** | **2.35T** | 74.4 |
| EVA-CLIP-8B (ours, distill) | **4.59B** | **2.35T** | **86.1** |
| DINOv2-g (Oquab et al., 2023) | 1.14B | 0.30T | **83.5** |
| DINOv2-g (ours, prune) | **0.62B** | **0.16T** | 76.4 |
| DINOv2-g (ours, distill) | **0.62B** | **0.16T** | **83.5** |

formance of compressed DINOv2-g model also reaches the one before pruning with only 54.4%

parameters of the original one. The above results exhibit the effectiveness of our method on pure vision transformer.

## 4.6 ABLATION STUDY

### 4.6.1 THE EFFECT OF INFORMATION ENTROPY CRITERION

To validate the effectiveness of our proposed criterion for neuron importance evaluation, we replace our proposed criterion with the popular criterion for Taylor pruning, cross entropy. For fair comparison, the parameters of model are reduced into roughly equal size. The results in Table 5 show that our proposed information entropy criterion is significantly superior to cross entropy on all zero-shot classification tasks. It indeed supports that our proposed criterion can provide more accurate importance scores for pruning, thus achieving higher performance after pruning.

Table 5: The effect of different criteria on zero-shot classification tasks.

| Criterion | #Params | IN-1K | IN-Adv | IN-R | IN-V2 | IN-Ske | ObjectNet | Avg. Acc. |
|---|---|---|---|---|---|---|---|---|
| Cross Entropy | 0.65B | 60.2% | 30.8% | 69.6% | 53.9% | 40.3% | 44.5% | 50.0% |
| **Information Entropy (ours)** | **0.62B** | **65.0%** | **34.0%** | **74.4%** | **57.7%** | **43.9%** | **47.9%** | **53.8%** |

### 4.6.2 THE EFFECT OF BINARY SEARCH

We compare our method with plain Taylor pruning with uniform pruning to analyze the effect of our proposed binary search strategy. For fairness, the hidden sizes of MLPs in the model pruned by plain Taylor pruning are reduced from 6144 to 2645, thus containing roughly equal parameter number as our pruned model. The results in Table 6 show that our method outperforms the plain Taylor pruning method by a significantly large margin, 64.5% average zero-shot classification accuracy on 6 benchmarks. It reveals that our method can adaptively reduce MLP modules in large vision transformer according to different redundancies.

Table 6: The effect of binary search on zero-shot classification tasks.

| Criterion | IN-1K | IN-Adv | IN-R | IN-V2 | IN-Ske | ObjectNet | Avg. Acc. |
|---|---|---|---|---|---|---|---|
| Taylor Pruning | 8.2% | 2.9% | 14.5% | 5.8% | 2.7% | 9.6% | 7.3% |
| **Binary Search (ours)** | **65.0%** | **34.0%** | **74.4%** | **57.7%** | **43.9%** | **47.9%** | **53.8%** |

### 4.6.3 THE EFFECT OF ENTROPY THRESHOLD

Furthermore, we analyze the effect of entropy threshold $\Delta\mathcal{E}$ for the termination of pruning. As shown in Table 7, we report the average classification accuracy of 6 zero-shot classification benchmarks, where the range of entropy threshold $\Delta\mathcal{E}$ is from $1 \times 10^{-4}$ to $1 \times 10^{-1}$. As expected, the parameter number of the pruned model is reduced with the increment of entropy threshold $\Delta\mathcal{E}$ from 0.73B to 0.43B and their corresponding average zero-shot accuracy is ranged from 62.5% to 31.3%. It demonstrates that a smaller increment in information entropy enables more discriminative capability of the pruned model, but also smaller parameter reduction.

Table 7: The effect of entropy threshold on zero-shot classification tasks.

| Threshold $\Delta\mathcal{E}$ | $1 \times 10^{-4}$ | $1 \times 10^{-3}$ | $5 \times 10^{-3}$ | $1 \times 10^{-2}$ | $2 \times 10^{-2}$ | $4 \times 10^{-2}$ | $1 \times 10^{-1}$ |
|---|---|---|---|---|---|---|---|
| #Params | 0.73B | 0.72B | 0.69B | 0.66B | 0.61B | 0.54B | 0.43B |
| Avg. Acc. | 62.5% | 61.5% | 57.6% | 53.8% | 48.5% | 41.7% | 31.3% |

## 5 CONCLUSION AND FUTURE WORK

In this paper, we study the compression of large vision transformers by the reduction of parameter-intensive MLP modules. To this end, we propose an Adaptive MLP Pruning method to significantly compress state-of-the-art large vision transformers in a near lossless manner. For more accurate neuron importance scores, we introduce a label-free information entropy criterion to fully model the prediction distribution during importance evaluation. Based on the obtained importance scores, we perform binary search algorithm to eliminate redundant hidden neurons of MLP modules in an adaptive fashion. Finally, we take the original model as the teacher to guide the pruned model for performance recover. Experimental results indicate that our method can substantially compress large vision transformers with only slight performance degradation. To further compress large vision transformer, we plan to explore the adaptive reduction of multi-head self-attention modules in future work. Moreover, we also expect to extend our method to the acceleration of large language model.

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

# A APPENDIX

## A.1 IMPLEMENTATION DETAILS

For information entropy based Taylor pruning, we set temperature coefficient $\tau$ and entropy threshold $\Delta\mathcal{E}$ as Table 8. For the knowledge distillation of our method, we adopt AdamW optimizer with bfloat16 precision. The learning rates of all models follows a cosine schedule for lr to min_lr, where lr = base_lr × batch_size / 256. We summarize batch_size, base_lr and min_lr in Table 8, where the setting of batch_size is based on the memory size of RTX A6000 GPUs (8 × 48G GPUs).

Table 8: The hyperparameters of our method. "DDP" and "FSDP" denote distributed data parallel strategy and fully sharded data parallel strategy, respectively.

| Model | dist. strategy | temperature $\tau$ | $\Delta\mathcal{E}$ | batch_size | base_lr | min_lr | weight decay | beta1 | beta2 |
|---|---|---|---|---|---|---|---|---|---|
| OpenCLIP-g | DDP | $\frac{1}{15}$ | 0.01 | 512 | $5 \times 10^{-5}$ | $1 \times 10^{-7}$ | 0 | 0.90 | 0.95 |
| OpenCLIP-G | DDP | $\frac{1}{15}$ | $2 \times 10^{-4}$ | 512 | $2 \times 10^{-5}$ | $1 \times 10^{-6}$ | 0 | 0.90 | 0.95 |
| EVA-CLIP-E | FSDP | $\frac{1}{20}$ | $2 \times 10^{-3}$ | 512 | $1 \times 10^{-5}$ | $1 \times 10^{-6}$ | 0 | 0.90 | 0.95 |
| EVA-CLIP-8B | FSDP | $\frac{1}{20}$ | 0.02 | 160 | $1 \times 10^{-5}$ | $1 \times 10^{-6}$ | 0 | 0.90 | 0.95 |
| DINOv2-g | DDP | $\frac{1}{20}$ | 0.01 | 512 | $7 \times 10^{-6}$ | $1 \times 10^{-7}$ | 0 | 0.90 | 0.95 |

## A.2 THE RELATION BETWEEN MLP HIDDEN SIZE AND INFORMATION ENTROPY

In this section, we reveal the relation between MLP hidden size and our proposed information entropy criterion on OpenCLIP-g model (the last 9 blocks, the original MLP hidden size is 6144). In this case, all hidden neurons of MLPs are ranked by importance scores in descending order. As shown in Figure 4, with the increment of MLP hidden size, the value of our proposed information entropy criterion monotonically decreases. In other words, the prediction uncertainty of model is reduced, when the MLP hidden size of model increases.

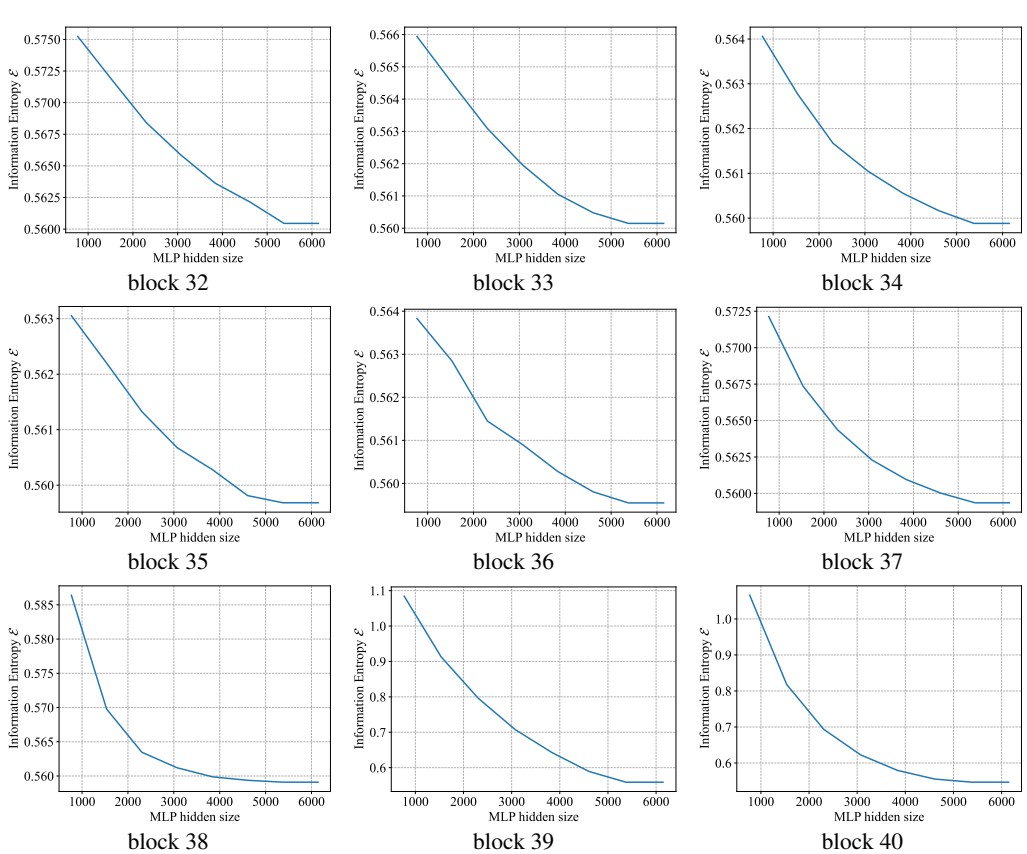

Figure 4: The relation between MLP hidden size and information entropy.

