# OpenReview forum: "Adaptive MLP Pruning for Large Vision Transformers"
_ICLR.cc/2026/Conference — ICLR 2026 Conference Withdrawn Submission_

### Official Review · Reviewer_TCWG · 2025-10-31

**Soundness:** 2
**Presentation:** 2
**Contribution:** 2
**Rating:** 4
**Confidence:** 5

**Summary:**

This paper proposes an Adaptive MLP Pruning (AMP) method to compress large vision transformers (ViTs) by reducing the number of neurons in their MLP modules, which are identified as the largest component of these models. The method introduces two key contributions. First, it proposes a novel label-free information entropy criterion for evaluating neuron importance. This criterion is based on the Taylor expansion of the model's output entropy, calculated from inter-instance similarity within a mini-batch. This approach avoids the limitations of traditional cross-entropy-based importance scores, which only consider the prediction for the ground-truth class, and it allows the method to be applied to models where the loss function or classification head is not public (e.g., DINOv2). Second, the paper introduces an adaptive pruning strategy that uses a binary search algorithm to determine the optimal number of neurons to prune for each MLP module individually. This avoids using a predefined, global compression ratio and allows the method to remove more redundant neurons from less important layers. Finally, knowledge distillation is used to recover the performance of the pruned model. The authors demonstrate through extensive experiments on large models like OpenCLIP, EVA-CLIP, and DINOv2 that AMP can reduce parameters and FLOPs by roughly 40% with near-lossless performance on zero-shot classification, retrieval, and kNN evaluation tasks.

**Strengths:**

- The use of a binary search to adaptively determine the pruning ratio for each layer is a major improvement over methods that rely on a fixed, global sparsity target. This allows the model to preserve capacity in more critical layers while aggressively pruning more redundant ones. The stark performance difference shown in Table 6 between AMP and uniform Taylor pruning (53.8% vs 7.3% average accuracy) convincingly demonstrates the effectiveness of this adaptive approach.
- The paper is backed by extensive experiments on a variety of modern, large-scale vision transformers (from 1B to 8B parameters). The evaluation is thorough, covering zero-shot classification, zero-shot retrieval, and kNN benchmarks. The results are impressive, consistently showing a ~40% reduction in parameters and FLOPs while recovering performance to be on par with, or even slightly better than, the original dense models after knowledge distillation.

**Weaknesses:**

- The proposed pruning method itself introduces a non-trivial computational cost. For each MLP module, the method performs a binary search for a fixed number of steps (t_max=6). In each step, it requires running forward passes on a dataset (D_prune of 50k images) to compute the information entropy. For very deep models with many MLP blocks, this iterative search process could be time-consuming. The paper does not analyze or report this search cost.
- The results show that performance drops dramatically after pruning and before distillation (e.g., from 73.0% to 53.8% for OpenCLIP-g in Table 1 and 3). The near-lossless results are entirely enabled by the subsequent, and costly, knowledge distillation step (10 epochs). While using KD for recovery is a common and valid practice, it's important to frame the contribution accurately: the method finds a good student architecture that can be effectively trained via KD, rather than being a "near-lossless" pruning method in itself.
- The method exclusively targets MLP modules. While the paper provides a strong justification that these modules contain the majority of parameters, it completely ignores other components like the multi-head self-attention (MHSA) blocks. A more holistic pruning approach that also considers attention heads or other parameters could potentially lead to better trade-offs between accuracy and efficiency. The authors acknowledge this as future work, but it remains a limitation of the current method.

**Questions:**

- Could you provide more details on the computational cost of the adaptive pruning search itself? Specifically, how much time does it take to determine the pruning ratios for a model like EVA-CLIP-E, and how does this cost compare to the 10-epoch knowledge distillation phase?
- The method introduces two key hyperparameters: the temperature τ for the similarity calculation and the entropy threshold ΔE for the binary search. Table 8 in the appendix shows that these values are set differently for each model. Could you elaborate on the process used to select these hyperparameters? How sensitive is the final pruned model's performance to their settings?
- The performance of the pruned models without distillation, while superior to other methods, is still quite low. Have you considered whether the excellent importance scores from your method could also benefit a simpler recovery process, such as standard fine-tuning on a labeled dataset (when available), instead of knowledge distillation? It would be interesting to see if a better pruning strategy leads to a better "warm-start" for fine-tuning as well.
- In the information entropy calculation (Eq. 5), you use the [CLS] token representation (z^cls) from the last transformer block. Have you experimented with using other representations, such as the average of all patch tokens, and did you observe any significant difference in the quality of the resulting importance scores?

---

> ### Author Response · Authors · 2025-11-20
>
> Thanks for your comments. We address your main concerns as below.
>
> - **How much time does it take to prune CLIP-E and 10-epoch knowledge distillation?**
>
>   For EVA-CLIP-E, we take 13.5 hours for pruning and 53.6 hours for 10-epoch knowledge distillation. Since our method **doesn't require the computation of parameter gradients and only require the computation of activation gradients**, importance evaluation of hidden neurons can be efficiently obtained.
>
> - **In our paper, we only claim near-lossless model compression / acceleration, instead of a "near-lossless" pruning method.** Moreover, our method also significantly outperforms other pruning methods without finetuning.
>
> - **How to select these hyper-parameters, temperature and entropy threshold?**
>
>   Due to different prediction distributions of different models, we empirically select proper temperature and entropy threshold to adjust their sensitivity on pruning.
>
>   For temperature, we adjust it to make different samples better distinguishing each other by information entropy value.  Then, we further select entropy threshold to balance performance and efficiency.
>
> - **The average of all patch tokens for information entropy calculation.**
>
>   We use average of all patch tokens for information entropy calculation and report the results on OpenCLIP-g as follows. All settings are consistent with the [CLS] token information entropy.
>
>   |       Method       | #Params | FLOPs |  Throughput  | IN-1K | IN-Adv | IN-R  | IN-V2 | IN-Ske | ObjectNet | Avg. Acc. |
>   | :----------------: | :-----: | :---: | :----------: | :---: | :----: | :---: | :---: | :----: | :-------: | :-------: |
>   |      original      |  1.01B  | 0.52T | 150.8 imgs/s | 78.5% | 60.8%  | 90.2% | 71.7% | 67.5%  |   69.2%   |   73.0%   |
>   |    [CLS], prune    |  0.62B  | 0.32T | 222.3 imgs/s | 65.0% | 34.0%  | 74.4% | 57.7% | 43.9%  |   47.9%   |   53.8%   |
>   |   [CLS], distill   |  0.62B  | 0.32T | 222.3 imgs/s | 78.4% | 60.8%  | 90.5% | 71.5% | 67.9%  |   69.4%   |   73.1%   |
>   |  avg patch, prune  |  0.66B  | 0.34T | 203.3 imgs/s | 62.1% | 31.3%  | 69.2% | 52.1% | 40.6%  |   42.6%   |   49.7%   |
>   | avg patch, distill |  0.66B  | 0.34T | 203.3 imgs/s | 78.1% | 60.2%  | 89.7% | 70.2% | 66.8%  |   68.9%   |   72.3%   |
>
>   The above results demonstrate that our [CLS] based information entropy is significantly better than the average patch token based one. We argue that [CLS] token is more discriminative than average patch one.

---

### Official Review · Reviewer_WHG1 · 2025-11-03

**Soundness:** 2
**Presentation:** 3
**Contribution:** 3
**Rating:** 6
**Confidence:** 3

**Summary:**

The paper introduces an adaptive MLP pruning approach for large vision transformers that replaces cross-entropy with information entropy to measure neuron importance. The method operates without labels or access to model-specific loss functions or heads. It uses a binary search procedure to determine pruning levels based on entropy changes. Experiments on several large models show significant parameter and FLOP reductions with limited accuracy degradation.

**Strengths:**

1. The experimental evaluation is extensive, spanning several large-scale vision transformers and diverse tasks, demonstrating consistent compression and performance trends.

2. The ablation studies are well organized and effectively disentangle the roles of the entropy criterion, pruning strategy, and threshold parameters.

3. The presentation is clear overall, with a logical flow and visual aids that make the method and procedure easy to follow.

4. The proposed approach is label-free and independent of model-specific heads or training losses, using an adaptive binary search to determine pruning ratios without manual tuning.

5. The pruning applies to neurons as units and achieve real-world speedup.

**Weaknesses:**

1. The evaluation is dominated by CLIP-style models, with only one experiment on DINOv2, leaving limited evidence of generalization beyond contrastive vision frameworks.

2. The work lacks experiments on standard supervised classification models, which makes it unclear how the method performs under typical vision transformer training settings.

3. The evaluation is confined to vision tasks, while recent pruning research increasingly focuses on large language models, limiting the relevance to broader transformer compression trends.

Part of the review is revised with LLM assistance.

**Questions:**

Please see weaknesses.

---

> ### Author Response · Authors · 2025-11-20
>
> Thanks for your constructive feedback. We handle your main concerns as follows.
>
> - **Experiments on standard supervised classification models, beyond contrastive vision frameworks.**
>
>   We conduct experiments on Swin transformer [1], which are trained on ImageNet-1k in a supervised manner. We set temperature $\tau=\frac{1}{20}$ and $\Delta \mathcal{E}=0.01$ for pruning and set batch size to 512, learning rate to $1\times10^{-4}$ for distillation. The results are listed as below.
>
>   |         Model          | #Params | FLOPs | Throughput  | Accuracy (%) |
>   | :--------------------: | :-----: | :---: | :---------: | :----------: |
>   |         Swin-B         |  87.8M  | 30.3G | 1652 imgs/s |     83.4     |
>   |  Swin-B (ours, prune)  |  47.2M  | 16.9G | 2735 imgs/s |     83.4     |
>   | Swin-B (ours, distill) |  47.2M  | 16.9G | 2735 imgs/s |     83.4     |
>
>   The above results support that the effectiveness of our method on supervised classification models, beyond contrastive vision frameworks.
>
>   [1] Liu et al. Swin transformer: Hierarchical vision transformer using shifted windows. CVPR 2021.
>
> - **Pruning on large language models.**
>
>   Thanks for your constructive suggestions.
>
>   Indeed, we are exploring the appliaction of our method on Large Language Models (LLMs). The inital results show that our method can be applied on LLMs. Our compressed LLaMA-3.2-1b achieves encouraging performance on several popular zero-shot language benchmarks, such as ARC-c and MMLU. However, there still exist some challenges to fully recover the original performance. To this end, we are refining the pruning solution based on our method to support data-efficient performance recover. We hope that we can share our research to machine learning community in the near future.

---

### Official Review · Reviewer_efLU · 2025-11-03

**Soundness:** 2
**Presentation:** 2
**Contribution:** 2
**Rating:** 2
**Confidence:** 5

**Summary:**

This paper proposed MLP pruning in vision transformer models based on an information entropy criterion. Specifically, to evaluate the neuron importance in MLP layers, this paper proposed to (1) calculate the pairs similarity of the output features of a batch of images, (2) define the output feature entropy based on the similarity of the query image to rest of images, (3) sum the entropy values up across the entire batch. Then the paper applied binary search to determine how many neuron to remove from each layer based on thresholding the information entropy difference before and after pruning. This method was applied to compress CLIP based vision transformer model and results on both classification and retrieval task are reported.

**Strengths:**

This paper is clearly written and well organized.

**Weaknesses:**

I have concerns in the following perspectives:
- The proposed criterion is essentially measuring the sample wise feature similarity among a batch of images. Larger entropy will mean more diverse output images. The reviewer is not sure why using feature similarity is a good criterion. How would pruning affect this information entropy value? Will it go up or down? Based on the formula $\epsilon_t - \epsilon_0 < \delta \epsilon$, it seems to indicate that the information entropy value will go up, meaning that removing neurons from MLP will make the model generate more diverse images? This seems contradict with classic neural network expressivity definition.
- Follow up question about the pruning criterion. The proposed criterion is label free, but there are other label free criterion that can be also applied in the Tylor pruning framework. For example, we can measure the model output feature reconstruction error like MSE before and after pruning and pick neurons with least impact to this reconstruction error as the criterion. The paper did not provide convincing comparison with this criterion in the ablation study.
- In the binary search algorithm, the paper actually adopted local pruning, i.e., rank and prune neurons within each layer, rather than global pruning. And the pruning process is conducted layer-by-layer. Although the ablation study showed that binary search is better than uniform pruning, but there is not comparison to global pruning which is supposed find more optimal pruning ratios among layers compared to layer-by-layer local pruning. For example, in layer-by-layer pruning, pruning first few layers can have higher tolerance since the rest of the layers are still intact. This means it is likely the final architecture becomes like a funnel shape. Can authors provide a visualization about the per-layer pruning rations after binary search?
- This paper only evaluated on the CLIP type vision transformers. The reviewer cannot understand why the authors only chose contrastive image-text based vision transformer models, rather than other types of vision foundation models such as DINO series, and even some of the well know VITs. I would recommend the authors extend the experiments to other vision transformer models, and include other vision tasks such as detection, segmentation, etc. The results so far cannot showcase the generality of the model.

**Questions:**

please refer to the weakness section.

---

> ### Author Response · Authors · 2025-11-20
>
> We address your main concerns as below.
>
> - **How would pruning affect this information entropy value?**
>
>   We have presented **the relation between MLP hidden size after and information entropy in Figure 4 (Section A.2 of appendix)**.
>
>   Figure 4 illustrates that, with the increment of MLP hidden size, the value of our proposed information entropy criterion monotonically decreases. In other words, the prediction uncertainty of model is reduced, when the MLP hidden size of model increases.
>
> - **Other criterions, e.g. MSE for feature reconstruction, can be used to Taylor pruning.**
>
>   **The MSE criterion is unable to work on Taylor pruning**. If MSE is adopted as criterion of Taylor importance evaluation, we can find that the **value and gradient of MSE** between unpruned model and the original model is **zero**, because **their prediction are always the same values**. In other words, **importance scores of all neurons in evaluated model are zeros**, according to the formula of Taylor pruning.
>
>   In contrast, our information criterion can provide valid non-zero gradient information for importance evaluation of Taylor pruning. Moreover, we don't require additional teacher model for the computation of criterion, as MSE criterion.
>
> - **Comparison to global pruning methods.**
>
>   We indeed report the comparative experiments on global pruning method, NViT, in Table 3 of Section 4.4. The results show that our method consistently outperforms global pruning method, NViT.
>
>   We summarize MLP hidden layer pruning ratios of our pruning model as follows.
>
>   - **OpenCLIP-g**
>
>     | Block 1-10 | 0.5704 | 0.1758 | 0.9915 | 0.1359 | 0.7237 | 0.1791 | 0.6520 | 0.8210 | 0.0289 | 0.0116 |
>     | :---------: | ---- | ---- | ---- | ---- | ---- | ---- | ---- | ---- | ---- | ---- |
>     | Block 11-20 | 0.5568 | 0.8482 | 0.2304 | 0.7266 | 0.5616 | 0.9485 | 0.2097 | 0.7214 | 0.2740 | 0.9325 |
>     | Block 21-30 | 0.4766 | 0.1142 | 0.4618 | 0.3013 | 0.2131 | 0.1126 | 0.8464 | 0.1132 | 0.9897 | 0.1523 |
>     | Block 31-40 | 0.8934 | 0.8332 | 0.8743 | 0.6991 | 0.1716 | 0.4770 | 0.2127 | 0.6329 | 0.4765 | 0.6367 |
>
>   - **OpenCLIP-G**
>   | Block 1-10 | 0.6125 | 0.3805 | 0.6399 | 0.7786 | 0.7791 | 0.2221 | 0.8125 | 0.0893 | 0.0823 | 0.7539 |
>   | :---------: | ---- | ---- | ---- | ---- | ---- | ---- | ---- | ---- | ---- | ---- |
>   | Block 11-20 | 0.3318 | 0.3014 | 0.8361 | 0.2961 | 0.7142 | 0.4144 | 0.7141 | 0.7860 | 0.4789 | 0.8364 |
>   | Block 21-30 |0.3857 | 0.8984 | 0.2515 | 0.9158 | 0.3276 | 0.5873 | 0.2709 | 0.8358 | 0.1439 | 0.9451 |
>   | Block 31-40 | 0.3892 | 0.0714 | 0.1633 | 0.9184 | 0.6980 | 0.9256 | 0.2054 | 0.7629 | 0.9190 | 0.1573 |
>   | Block 41-48 |0.0713 | 0.3762 | 0.7414 | 0.0929 | 0.3248 | 0.4460 | 0.5736 | 0.5101 |  |
>
>
>   - **EVA-CLIP-E**
>   | Block 1-10 | 0.5124 |  0.5157 |  0.3002 |  0.3500 |  0.4843 |  0.4585 |  0.4159 |  0.4885 |  0.3197 |  0.5347 |
>   | :---------: | ---- | ---- | ---- | ---- | ---- | ---- | ---- | ---- | ---- | ---- |
>   | Block 11-20 |0.4355 |  0.0170 |  0.3030 |  0.4323 |  0.0037 |  0.4623 |  0.4922 |  0.1415 |  0.4133 |  0.3248 |
>   | Block 21-30 |0.2361 |  0.4354 |  0.0212 |  0.3581 |  0.1940 |  0.5246 |  0.5822 |  0.2676 |  0.3050 |  0.0990 |
>   | Block 31-40 |0.1707 |  0.5167 |  0.5487 |  0.3448 |  0.4233 |  0.5595 |  0.0419 |  0.0869 |  0.3378 |  0.2097 |
>   | Block 41-50 |0.1851 |  0.1304 |  0.0181 |  0.1529 |  0.2322 |  0.1119 |  0.4590 |  0.3410 |  0.4827 |  0.0998 |
>   | Block 51-60 |0.2662 |  0.4034 |  0.5276 |  0.3126 |  0.5954 |  0.2869 |  0.2670 |  0.5396 |  0.1934 |  0.3066 |
>   | Block 61-64 |0.0099 |  0.0800 |  0.3428 |  0.0418 | | | | | | |
>
>
> - **Evaluation on non-CLIP-type model.**
>
>   We indeed evaluate our method on non-CLIP-type large vision transformer, **DINOv2-g in Table 4 of Section 4.5**. Our method recovers kNN accuracy of the original model with only 54% parameters.

---

### Official Review · Reviewer_Gfz2 · 2025-11-09

**Soundness:** 2
**Presentation:** 3
**Contribution:** 2
**Rating:** 4
**Confidence:** 3

**Summary:**

The paper shows a method to prune parameters of hidden layers in VLMs to reduce the memory consumption of the model and the computation of inference. The author proposed to use label agnostic neuron scoring during taylor ranking, instead of label-specific information obtained from standard taylor method to more accurately capture the neuron importance. Then a binary search is performed on the taylor importance to determine the pruned neuron. The authors provided results on a few CLIP based models, as well as conducted extensive ablation discussion.

**Strengths:**

The paper well written, the presentation of the idea is clear.

**Weaknesses:**

The novelty seems limited, where the core idea is to replace standard cross entropy-based gradient used in taylor-based importance score with information entropy which is label-agnostic, and the rest of it which is binary search become quite obvious. The method proposed therefore seems simple and straightforward, which then requires comprehensive experiments to justify the generalizability of it. However, the experiment discussions are not very comprehensive.

**Questions:**

1. Regarding the information entropy-based criterion which seems unsupervised (require no label information), compared to supervised criteria, i wonder whether there are some cases will mislead the pruning method, e.g. encouraging a pruned structure that is overconfident on wrong prediction.
2. Comparisons with other pruning methods are limited. Baseline methods selected for comparions in Table 3 are not enough and slightly outdated.
3. The method is only evaluated on the CLIP-based models with up to 4.59B. I wonder how it will perform on more mainstream models like Llava, Phi-V, Qwen-VL, etc. The two models used for lateral comparisons are also smaller variants in their respective model series. Whether the performance improvements are consistent on those larger models remains.
4. The evaluation would be more comprehensive if there are some hardware speedup and memory reduction benchmarks to support this method.

---

> ### Author Response · Authors · 2025-11-20
>
> Thanks for your helpful comments. We address your main concerns as follows.
>
> 1. The main target of our method is **to preserve the capability of the original model after pruning**, **instead of the error correction of the original model**. The experimental results indeed support that our method can well recover the performance of the original model without annotations, well supporting the main contribution of our paper.
> 2. Some pruning methods are **too time-consuming to prune large vision models**. Hence, in this paper, we focus on methods, which are more efficient implemented on large vision model.
> 3. As we know, **vision encoders** adopted in Llava, Phi-V and Qwen-VL are also **CLIP-like models**, e.g. ViT-Large for Llava, whose parameters are significantly fewer than 7.53B of EVA-CLIP-8B. The main contribution of our paper **focuses on the compression of vision encoder**. Of course, the compression of LLM parts is another interesting topic, but out of scope of this paper.
> 4. Indeed, we **have reported hardware speedup** in **Table 1 of our paper**. We evaluate **image throughput** of the pruned models on single A6000 GPU with batch size of 1000. For memory reduction of large vision model, since model parameters take the most memory footprint, its memory reduction can be roughly obtained form its parameter reduction.

---

### Note · Authors · 2026-02-05

I have read and agree with the venue's withdrawal policy on behalf of myself and my co-authors.

---

### Meta-Review · Area_Chair_S2o1 · 2026-01-06

**Summary:**

The main decision‑informing concerns raised by the reviewers are insufficient novelty and a lack of thorough comparisons with baseline methods. The limited application to vision transformers is also an issue, although, as one reviewer stated, “the idea of replacing the standard cross‑entropy-based gradient used in Taylor‑based importance scoring with label‑agnostic information entropy” appears more general.

**Reviewer Concerns:**

Some reviewer concerns are only partially addressed. For example, Reviewer Gfz2’s comments about the limited novelty have not been tackled in the rebuttal. Comments on the experiments "Comparisons with other pruning methods are limited. Baseline methods selected for comparisons in Table 3 are not enough and slightly outdated" have not been well addressed. The authors only compared methods that are efficiently implementable on large vision models, but they did not provide execution‑time criteria or analyses of the methods they chose not to compare.

Reviewer WHG1 raised concerns that the application scope of the proposed method is narrow: its evaluation is confined to vision tasks, while recent pruning research increasingly focuses on LLMs, limiting its relevance to broader transformer compression trends. In the rebuttal, the authors expressed that they are exploring the application of their method to LLMs. The initial results suggest that the method can be applied to LLMs, but failed to fully recover the original performance yet.

**Reviewer Scores:**

I think most reviewers will maintain the original scores. The authors didn't provide the paper-revision information in the rebuttal.

---

### Decision · Program_Chairs · 2026-01-26

Reject